# Both Medical and Context Elements Influence the Decision-Making Processes of Pediatricians

**DOI:** 10.3390/children9030403

**Published:** 2022-03-11

**Authors:** Lisa Schurmans, David De Coninck, Birgitte Schoenmakers, Peter de Winter, Jaan Toelen

**Affiliations:** 1Faculty of Medicine, KU Leuven, 3000 Leuven, Belgium; lisa.schurmans@student.kuleuven.be; 2Centre for Sociological Research, KU Leuven, 3000 Leuven, Belgium; 3Leuven Child and Youth Institute, KU Leuven, 3000 Leuven, Belgium; pdewinter@spaarneziekenhuis.nl (P.d.W.); jaan.toelen@uzleuven.be (J.T.); 4Department of Public Health and Primary Care, KU Leuven, 3000 Leuven, Belgium; birgitte.schoenmakers@kuleuven.be; 5Department of Development and Regeneration, KU Leuven, 3000 Leuven, Belgium; 6Department of Pediatrics, Spaarne Gasthuis, 2134 TM Hoofddorp, The Netherlands; 7Department of Pediatrics, KU Leuven, 3000 Leuven, Belgium

**Keywords:** medical decision-making, pediatrics, contextual factors

## Abstract

We wanted to investigate the relationship of medical and non-medical factors with the clinical decision-making of pediatricians. We hypothesize that the addition of relevant medical information (either alarming or reassuring) will influence the physician’s decision-making, but that the addition of non-medical information will also play a role. To investigate this, we designed an online questionnaire containing ten clinical case-based scenarios, of which five focused on medical factors and five on non-medical/context factors, each scored on a five-point Likert scale. In total, 113 pediatricians completed the online questionnaire. Both medical and non-medical/context factors were considered relevant to change the initial decision in most cases. Additional information of an alarming nature induces the physician to become more worried, whereas reassuring information decreases this worry. In some cases, with the medical factors, the gender and the age of the pediatrician does have some effect on the clinical decision-making. We conclude that medical decision-making is affected by multiple intrinsic and extrinsic factors that differ between physicians. Our data indicate that these non-medical factors must be considered when making a medical decision, as it is crucial to be aware that they have a substantial influence on that decision-making.

## 1. Introduction

Clinical reasoning and medical decision-making are two crucial skills of any physician. The groundwork for this decision-making usually lies in physicians’ training, where basic medical concepts which are often illustrated with real life clinical cases. Subsequently, physicians also acquire experience based on their exposure to numerous and diverse cases, combined with advice from fellow physicians. Gradually, this experience is organized into clinical scripts (or illness scripts) [1], which are used to efficiently recognize a constellation of symptoms in a patient throughout their professional life. Based on this theory one could erroneously conclude that clinical reasoning is nothing more than the application of a neurophysiological decision algorithm. Instead, it is a far more complex activity in which analytical and non-analytical reasoning play a significant role [2,3].

The cognitive processes that underlie the clinical reasoning can be explained by the Dual Process Theory which holds that physicians use two forms of reasoning [4,5,6]. The first form of reasoning consists of instant pattern recognition (i.e., non-analytic reasoning or a form of clinical ‘Gestalt’ recognition). It is an intuitive, automatic, fast-processing system that can almost be considered subconscious. This reasoning includes the use of heuristics (mental shortcuts) to solve problems, which may lead to cognitive biases and even result in medical errors. On the other hand, there is a more analytical, effortful, slower type of reasoning. This is more self-aware and reflective. It is primarily used in more complex situations and will likely lead to less error [7,8]. An experienced physician will combine these two types of reasoning when confronted with clinical problems, whereas a physician in training will often use analytic reasoning. As mentioned above, using these heuristics is faster but comes at a cost as the process is more prone to other influences (sleep deprivation, overstimulation, hunger, affect…), which may lead in some cases to faulty reasoning and medical errors. On the other hand, more experienced physicians gradually encounter more exceptions or non-classical presentations of clinical scenarios, making them more cautious [7].

Recent studies show that not only objective medical information but also non-medical elements can influence clinical decision-making [9,10,11,12,13]. ‘Non-medical’ elements can be defined as factors that appear to have no direct or objective clinical or medical relevance to the patient’s disease or situation. These can include elements such as the psychological or socio-cultural characteristics of both the physician and the patient or the specific context in which the decision-making occurs. For example, the gut feeling of a colleague, the worrisome character of a mother, a similar case in the past but with a negative outcome and the physician’s personal medical history. At the sociological level, elements such as the socio-economic status of patients but also their religious denomination and cultural preferences can have an impact [14,15].

This study aims to investigate the relationship of medical and non-medical elements with the clinical decision-making of pediatricians. We used fictional but realistic clinical cases within a pediatric setting to assess this link that led to an ‘initial diagnosis’. Subsequently, we provided each case with an additional specific medical or non-medical element. We hypothesize that the addition of relevant medical information (be it alarming or reassuring) will influence the physician’s decision-making, but that the addition of non-medical information will also play a role in this process.

## 2. Materials and Methods

### 2.1. Study Design

To gather cross-sectional data, we developed an online questionnaire, which contained four demographic questions and 10 clinical case-based scenarios (that involve a decision-making process of a pediatrician), of which five focused on medical factors and five on non-medical or context factors. We asked about respondents’ gender, age, years of experience, and the number of biological children in terms of demographic information.

The five clinical scenarios with medical factors included fever with shivering, the localization of petechiae in the face, paresis of the arm after febrile convulsions, iron-deficiency anemia with a palpable spleen and changed stool pattern with anal blood loss. The contextual, non-medical factors related to the gut feeling of a general practitioner, the patient’s mother’s profession as a nurse, the patient’s parents’ unkempt appearance, a distraught mother, and a past negative experience by a colleague with a similar case. The replies on these scenarios were scored on a five-point Likert scale, from 1 = ‘much less likely to run more tests’ to 5 = ‘much more likely to run more tests’. For a complete overview of all cases, see Appendix A.

### 2.2. Setting and Subjects

The questionnaire was programmed on Qualtrics, an online tool to develop and program surveys. It was subsequently emailed to 130 members of the Flemish Association for Pediatricians (professional association) and to all pediatricians in training at KU Leuven (university). We focused on pediatricians as they often face complicated cases given the inclusion of parents in the medical decision-making process for minors. Thus, we expected contextual factors (e.g., related to the patient’s parents) to play a particularly important role here, although this will certainly also be the case for physicians in other fields. The survey was fielded from early February 2021 until the end of April 2021, at which point we had 116 participants. An information letter was attached to the email and included at the start of the survey. It was clearly stated that participation was voluntary, that data collection was completely anonymous and that by continuing, the participant agreed with the informed consent. This study was approved by the KU Leuven Research Ethics Committee (case number MP016831) on 15 December 2020.

### 2.3. Data Analysis

Quantitative, descriptive data analysis regarding demographic data and Likert scale questions were conducted using IBM SPSS Statistics version 27. In the first step, we provide a descriptive overview of our study sample. Subsequently, we present the results of the answers to the ten clinical case-based scenarios. Finally, we conducted 10 linear regression analyses (one per case) to assess which factors (age, gender, years of experience, number of biological children) are linked to pediatricians’ decision-making in each case.

Research manuscripts reporting large datasets that are deposited in a publicly available database should specify where the data have been deposited and provide the relevant accession numbers. If the accession numbers have not yet been obtained at the time of submission, please state that they will be provided during review. They must be provided prior to publication.

## 3. Results

### 3.1. Sociodemographic Characteristics

Out of the 116 respondents, 113 completed the questionnaire. Of these 113 participants, nearly three quarters were women (73.5%). The median age of the respondents was 36 years old, with a minimum age of 25 years and a maximum age of 72 years. In terms of years of experience, we note that 53.1% has under five years of experience (and thus, are still in training). Over half of the participants have children of their own (57.5%). The different socio-demographic characteristics of the participants are detailed in Table 1.

### 3.2. Responses to Case-Based Scenarios

As seen in Figure 1 and Figure 2, the additional medical factors in most cases, with the exception of casus 1, were overall considered relevant to change the initial decision and run more tests. In the medical cases about paresis of the arm, palpable spleen and rectal blood loss, the majority of the answers ranged from “more likely to run more tests” to “very more likely to run more tests”, similar to the answers from case 2 being “less likely to run more tests” to “very less likely to run more tests”.

As for all the cases with non-medical factors, the majority of the participants will change their decision slightly or not at all, with the overall answers ranging from “changes nothing” to “less likely to run more tests” and “more likely to run more tests”. As seen in case 10, the experience of a colleague has more impact on female pediatricians and those in training. Additional analysis can be found in Appendix B (Table A1).

Note: ‘Gut feeling’ relates to the gut feeling of the general practitioner; ‘mother is a nurse’ is on the patient’s mother’s profession as a nurse, ‘unkempt parents’ is on the patient’s parents’ unkempt appearance, ‘worried mother’ is about a distraught mother, and ‘experienced colleague’ deals with a past negative experience by a colleague with a similar case.

### 3.3. Regression Analysis with Case-Based Scenarios as Dependent Variables

Adjusted R-squared reflects the percentual predictive value the independent variables have together for each case. The most significant results can be seen in the cases with an additional medical factor (Table 2). As seen in case 5, the contribution of the independent variables is 16.7% (adj. R sq = 0.167). The age, years of experience and gender are significant (*p* < 0.01). In case 1, the independent variables contribute for almost 10% (adj. R sq = 0.096). Here, age (*p* < 0.01) and years of experience (*p* < 0.05) are also significant. As for case 3, only the gender is significant (*p* < 0.05).

As for the non-medical cases, for example, case 8, the total contribution of the independent variables is 9% (adj. R sq = 0.092). Here, the years of experience (*p* < 0.01), age (*p* < 0.05) and gender (*p* < 0.05) are the most significant. In case 10 the contribution is 14% (adj. R sq = 0.139), but there are no significant factors. When we consider the overall mean score of the five medical and five non-medical cases, results indicate that gender was a decisive factor: female pediatricians were significantly more likely to request further testing in both medical (*β* = 24, *p* < 0.05) and non-medical cases (*β* = 0.25, *p* = 0.05) than male pediatricians.

## 4. Discussion

This study investigated which factors are associated with decision-making given either medical or non-medical/context information. We also looked at whether the likelihood of requesting additional testing differs between medical and non-medical cases. Findings indicate that medical factors were—four out of five cases—strongly linked with medical decision-making among pediatricians. Although changes in decision-making by pediatricians was somewhat less pronounced for contextual, non-medical factors overall. Here, we note differences depending on the particular type of contextual factor under consideration. For example, the role of the experience of a colleague appears to play a greater role than factors related to the patient’s mother. Regression analyses indicate that pediatricians’ age and gender are linked with medical decision-making.

### 4.1. Medical Factors

Medical factors have an apparent effect on the clinical decision-making process of pediatricians in training as well as experienced ones. Additional information of an alarming nature induces the physician to become more worried about a case, whereas reassuring information decreases this worry.

More specifically, we notice that in the clinical cases with additional alarming medical information (i.e., a palpable spleen, anal blood loss and paresis of the arm after febrile seizures), most of the pediatricians are ‘more likely’ and ‘much more likely’ to request more additional investigations or tests. This result is as expected because the additional information given in these cases suggests a more atypical, complex or severe condition [16,17,18,19].

This effect is also present for the symptom of shivering during a febrile episode, but to a lesser extent. In the past shivering during a febrile episode was considered a significant sign, suggestive of a more serious infection (before the advent of immunizations for streptococcal and meningococcal infections). Yet a recent meta-analysis suggests that this is no longer the case [20]. It may be that this recent evidence has not yet reached all our study participants, many of whom would ‘more likely’ run more tests. In this case we would have anticipated that an up-to-date physician would not have changed his/her approach.

In the case of the child with fever, vomiting and petechiae the initial diagnosis was that of invasive infection. Yet the provision of reassuring information, i.e., the localization of the petechiae were in the face (in a child that has been vomiting), changed the pediatricians’ reasoning towards a more restricted management [21,22,23].

### 4.2. Non-Medical Factors

It has been suggested before that clinical reasoning may not only be symptom or case-specific, but also context-specific. The concept of ‘context specificity’ refers to the observation that a doctor can see two patients with the same symptom or clinical complaint or with similar physical findings but—in different contexts—end up with different diagnoses [24]. This context can either help the physician make the correct diagnosis or lead to diagnostic or medical error [25]. This suggests that factors other than the “essential medical content” is influencing the doctor’s clinical reasoning. We reasoned that these ‘non-medical factors’ (or context factors) would also have an effect on the clinical decision-making of the participants in our study. We based the factors that were the subject of these cases on published data about non-clinical influences [13,26].

We used different context factors in our cases, some regarding the parents (unkempt appearance, worried mother and the medical training of a mother), some regarding colleagues (gut feeling of a referring physician and advice of an experienced colleague). In the case of the child with gastroenteritis with mild dehydration, the parents’ unkempt appearance affected the pediatricians’ response. During an encounter in the clinic or in the consultation room, a physician will try to assess the knowledge, attitudes, and practices (especially those associated with positive child-parent interactions and promote the child’s healthy development) of parents who accompany a child. Such context factors (e.g., language barrier, low education level, or other sociological factors such as low socio-economic situation or religious denomination…) may act as barriers to an efficient parent-physician interaction and influence the medical decision-making process [27].

Our data show that the ‘gut feeling’ of a general practitioner or the advice of an experienced colleague influences the decision-making of pediatricians in some cases. The gut feeling of physicians is a well-studied area and its significance in daily practice is established [28,29]. It is clear that these factors have a major impact on the clinical decision-making of physicians, 55–60% of the participants answer that they are more likely to change their initial diagnosis based on this information.

Pediatricians generally believe that a worried mother should always be taken seriously (until proven otherwise) [30,31]. In the case of the child with the tension-type headache, 45% of pediatricians would change their initial assessment and run more tests due to the mother’s worries. It is known that maternal anxiety is a context factor that influences the clinical reasoning of a physician [32].

In the case of the child with the concussion, the context that the mother is a trained nurse reassures many pediatricians and changes their management of the case from ‘admission for neurological observation’ to ‘discharge with observation at home’. It is clear from the medical information of the case that this is not a major craniocerebral injury, but according to the PECARN guideline, it would still warrant observation or neuroimaging [33]. One could argue that the mother may be capable of doing this at home, on the other hand, a hospital may be a better environment to cope with the occurrence of complications such as somnolence, vomiting or even more severe adverse events.

### 4.3. Regression Analysis

In some cases, with the medical factors, the gender and the age of the pediatrician does have some effect on the clinical decision-making. Women tend to adjust their decision more than their male colleagues. We could not find data to compare this finding in the current medical literature. The number of children does not seem to influence decision-making in these cases.

### 4.4. Methodological Considerations

We used hypothetical scenarios constructed as a modified script concordance test to probe the personal judgment that is made in the clinical reasoning process. In doing so we forced the participants to start their clinical reasoning from an initial diagnosis and investigated what the effect of additional information was. This is a fair approximation, but it is not a perfect substitute for the reasoning process in the clinic where numerous other factors may also influence the final decision. The fact that the survey was voluntary could have led to selection bias in our study cohort. In our analysis we dichotomized the participant group into experienced or not based on their years in training. While the change into a more experienced physician is a gradual process, we hypothesized that a physician in training would have a lower-case repertoire. Their overall reasoning could also be influenced by the fact that they still regularly discuss cases with their tutors.

## 5. Conclusions

Medical decision-making is not a process that can be translated into an infallible algorithm. It is a process that is affected by multiple intrinsic and extrinsic factors that differ between physicians. In this study we have shown that both clinical as well as contextual factors influence the process of medical decision-making in pediatricians.

## Figures and Tables

**Figure 1 children-09-00403-f001:**
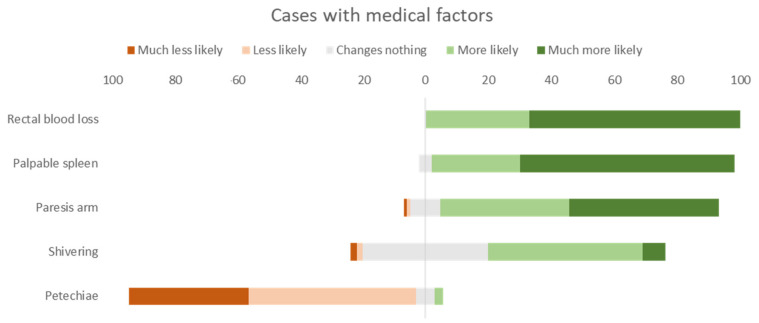
Overall answers to cases with medical factors in percentages.

**Figure 2 children-09-00403-f002:**
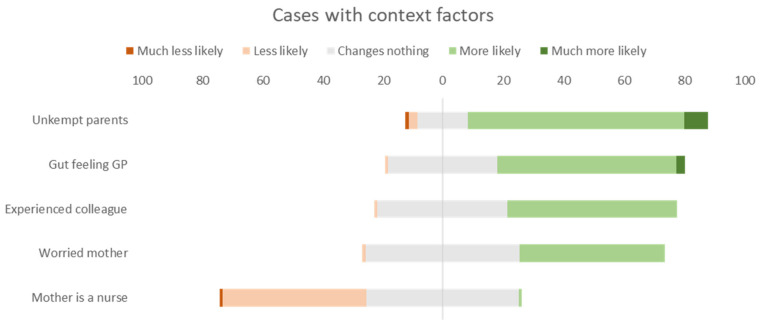
Overall answers to cases with non-medical factors in percentages.

**Table 1 children-09-00403-t001:** Socio-demographic characteristics of participants (*N* = 113).

Participant Characteristics	*N* (Percentages)
Gender	
Female	83 (73.5)
Male	30 (26.5)
Age	
<30	40 (37.3)
30–39	35 (32.7)
40–49	16 (15.0)
50 or older	16 (15.0)
Years of experience	
Less than 5	60 (53.1)
5–10	17 (15.0)
10–15	11 (9.7)
15–20	6 (5.3)
20 or more	19 (16.8)
Number of children	
None	48 (42.5)
1	15 (13.3)
2	21 (18.6)
3	24 (21.2)
4	4 (3.5)
5 or more	1 (0.9)

**Table 2 children-09-00403-t002:** Linear regressions with case-based scenarios as dependent variables.

	Case 1	Case 2	Case 3	Case 4	Case 5	Case 6	Case 7	Case 8	Case 9	Case 10	Medical Factors	Non- Medical Factors
Gender (ref: male)												
Female	0.17 (0.16)	−0.13 (0.16)	0.24 * (0.17)	0.08 (0.13)	0.34 ** (0.10)	−0.01 (0.13)	0.11 (0.12)	0.20 (0.15)	0.09 (0.12)	0.19 (0.11)	0.24 * (0.09)	0.25 * (0.06)
Age	−0.80 ** (0.02)	0.53 (0.02)	−0.36 (0.02)	−0.27 (0.01)	0.92 ** (0.01)	0.14 (0.02)	0.57 (0.02)	−0.57 * (0.02)	0.28 (0.01)	−0.25 (0.01)	−0.38 (0.01)	0.01 (0.01)
Years of experience	0.60 * (0.13)	−0.44 (0.13)	0.10 (0.14)	0.16 (0.10)	−0.77 ** (0.08)	−0.21 (0.11)	−0.56 (0.10)	0.75 ** (0.12)	−0.12 (0.10)	−0.01 (0.09)	0.21 (0.07)	0.02 (0.05)
Number of children	0.17 (0.07)	−0.03 (0.07)	0.21 (0.08)	0.15 (0.06)	−0.25 (0.04)	0.03 (0.06)	0.17 (0.05)	−0.14 (0.06)	0.10 (0.05)	−0.10 (0.05)	0.22 (0.04)	0.01 (0.03)
Adjusted R²	0.10	0.02	0.08	0.01	0.17	0.01	0.04	0.09	0.02	0.14	0.07	0.02

Note: ** *p* < 0.01; * *p* < 0.05. Standardized coefficients presented, standard error between brackets. High score = more likely to request further testing.

## Data Availability

The data presented in this study are available on request from the corresponding author. The data are not publicly available due to restrictions regarding participants’ identity.

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
