# Peer review of "Both Medical and Context Elements Influence the Decision-Making Processes of Pediatricians"

_children, 2022, doi:10.3390/children9030403_

Round 1

Reviewer 1 Report

In the manuscript # children-1601577, Authors through an online survey on 10 clinical vignettes investigated medical and, especially, non-medical factors able to affect the clinical decision-making on whether to adopt additional investigation. I thank the Author for this work, I enjoyed reading it.

Specific observations

  • Invitation was sent to 130 members of the Flemish Association for 90 Pediatricians (professional association) and to all pediatricians in training at KU Leuven (university). No potential respondents were here described. This would apply to responses’ weighting by non-respondents’ fractions, but this adjustment was not adopted [JAMA Surg. 2020;155(4):351–352.]. Please describe whether this approach can be applied or not, and eventual causes of dropping such methodology.
  • While the five clinical cases are readily understandable from Figure 1, the five questions regarding non-medical factors should be more clearly explained in the text, at least in the footnote of the Figure 2
  • Please report how the overall score of medical and non-medical cases were counted (the simple sum, I suppose)
  • Hope that Authors forgot to remove from the text some template suggestions, as This section may be divided by subheadings. It should provide a concise and precise 112 description of the experimental results, their interpretation, as well as the experimental 113 conclusions that can be drawn” Reported in the Results section.
  • The first part of introduction, about medical school, seems not pertinent to the aim of the study and should be removed. The same applies when discussing about analytic reasoning in trainees. I understood that age was found as a predictor, and that some respondents were residents, but this is a post-hoc observation, not included in the a-priori hypothesis.

Author Response

In the manuscript # children-1601577, Authors through an online survey on 10 clinical vignettes investigated medical and, especially, non-medical factors able to affect the clinical decision-making on whether to adopt additional investigation. I thank the Author for this work, I enjoyed reading it.

Comment 1: Invitation was sent to 130 members of the Flemish Association for 90 Pediatricians (professional association) and to all pediatricians in training at KU Leuven (university). No potential respondents were here described. This would apply to responses’ weighting by non-respondents’ fractions, but this adjustment was not adopted [JAMA Surg. 2020;155(4):351–352.]. Please describe whether this approach can be applied or not, and eventual causes of dropping such methodology.

Response: Although we acknowledge that weighting may be a useful method when attempting to recreate a representative sample of a certain population and reduce potential sources of bias, it was not necessarily our goal to do so. This study serves as a first exploration into medical decision-making based on contextual elements, and as such, we did not attempt to create a dataset that was representative. This would be impossible with regards to pediatricians in training in Flanders given that various other Flemish universities also train pediatricians, but data was not collected there. We also acknowledge and reflect on this in section 4.4.

Comment 2: While the five clinical cases are readily understandable from Figure 1, the five questions regarding non-medical factors should be more clearly explained in the text, at least in the footnote of the Figure 2.

Response: We have expanded some more on the explanation of the cases under heading 2.1. We have also added a note under Figure 2 that briefly explains each case.

Comment 3: Please report how the overall score of medical and non-medical cases were counted (the simple sum, I suppose).

Response: We have clarified in the text that we meant the mean score for each ‘type’ of case.

Comment 4: Hope that Authors forgot to remove from the text some template suggestions, as “This section may be divided by subheadings. It should provide a concise and precise 112 description of the experimental results, their interpretation, as well as the experimental 113 conclusions that can be drawn” Reported in the Results section.

Response: This was indeed an oversight in our part, thank you for noticing this. We have removed this.

Comment 5: The first part of introduction, about medical school, seems not pertinent to the aim of the study and should be removed. The same applies when discussing about analytic reasoning in trainees. I understood that age was found as a predictor, and that some respondents were residents, but this is a post-hoc observation, not included in the a-priori hypothesis.

Response: We agree that the first paragraph focused too much on the training aspect of physicians. We have revised the first paragraph and moved away from this perspective. The same goes for the paragraph regarding the analytic reasoning in trainees in the discussion.

Reviewer 2 Report

It is an interesting and important manuscript to assess both biomedical elements and the social context in taking decision-making processes in pediatrics.

I think some aspects of expanding on in the study

Introduction

Although there are few quotes related to social elements that influence the decision-making processes, it is well written.

I suggest adding a paragraph in this regard.

Material and methods

Describe the type of study design. Is it a cross-sectional study? The research strategy is described. Was the questionnaire developed validated? What was the reason for selecting these cases?

Describe the elements of the social context since the description emphasizes the medical elements that influence decision-making and some psychological but not social elements. It is essential to describe it because, in the discussion, you are focus a lot on them and the fact that these elements would be the most novel part of the study.

Results

They are well presented and in a clear form.

Discussion

The first paragraph of the discussion should summarize the most relevant findings. I suggest adding it.

The discussion of non-medical factors focuses on psychological elements but not social.

The biases of the study of the participation of more women pediatricians Does it correspond to the distribution of pediatricians of the Flemish Association and KU Leuven?

Conclusion

It does not correspond to the study. What is the direct conclusion or conclusions of the study?

What is presented are reflections of the authors.

Author Response

It is an interesting and important manuscript to assess both biomedical elements and the social context in taking decision-making processes in pediatrics.

I think some aspects of expanding on in the study

Introduction

Comment 1: Although there are few quotes related to social elements that influence the decision-making processes, it is well written.

I suggest adding a paragraph in this regard.

Response: We have added an additional paragraph relating to the social elements that may influence decision making in the introduction with additional references.

Material and methods

Comment 2: Describe the type of study design. Is it a cross-sectional study? The research strategy is described. Was the questionnaire developed validated? What was the reason for selecting these cases?

Response: The study was cross-sectional, as is now described under heading 2.1. Under heading 2.2, we now added a brief motivation for why we selected these specific cases. The cases are based on concepts that have been described in the literature before, but rather than asking physicians in general about contextual or clinical factors that influence their reasoning, we opted for a case based approach, which is much closer to the clinical reality than a more ‘abstract’ questionnaire.  

“We focused on pediatricians as they often face complicated cases given the inclusion of parents in the medical decision-making process for minors. Thus, we expected contextual factors (e.g., related to the patient’s parents) to play a particularly important role here, although this will certainly also be the case for physicians in other fields.”

Comment 3: Describe the elements of the social context since the description emphasizes the medical elements that influence decision-making and some psychological but not social elements. It is essential to describe it because, in the discussion, you are focus a lot on them and the fact that these elements would be the most novel part of the study.

Response: We have added additional text relating to the social factors both in the introduction and the discussion of our manuscript (with the relevant references as mentioned above).

Results

They are well presented and in a clear form.

Discussion

Comment 4: The first paragraph of the discussion should summarize the most relevant findings. I suggest adding it.

Response: We agree with your assessment, and have added an additional paragraph at the beginning of the discussion:

Findings indicate that medical factors were – four out of five cases – strongly linked with medical decision-making among pediatricians. Although changes in decision-making by pediatricians was somewhat less pronounced for contextual, non-medical factors overall. Here, we note differences depending on the particular type of contextual factor under con-sideration. For example, the role of the experience of a colleague appears to play a greater role than factors related to the patient’s mother. Regression analyses indicate that pediatricians’ age and gender are linked with medical decision-making.

Comment 5: The discussion of non-medical factors focuses on psychological elements but not social.

Response: As mentioned above we have accounted for this in the new version of the manuscript.

Comment 6: The biases of the study of the participation of more women pediatricians Does it correspond to the distribution of pediatricians of the Flemish Association and KU Leuven?

Response: We have asked the official instances for the data regarding the number of female pediatricians in Flanders and also looked at the gender ratio on trainees at KU Leuven. This ratio is 70%/30% Female/Male (in 2020, most recent data), which is in line with the number of women who responded to our survey. Thus there does not seem to be an overrepresentation of female participants in this study.

 Conclusion

Comment 7: It does not correspond to the study. What is the direct conclusion or conclusions of the study? What is presented are reflections of the authors.

Response: We have changed the conclusion to be more in line with our findings.

Round 2

Reviewer 2 Report

The authors had changes based on all recommendation. The manuscript are suitable to publish.